# Boosting of cross-reactive antibodies to endemic coronaviruses by SARS-CoV-2 infection but not vaccination with stabilized spike

Andrew R Crowley[1†], Harini Natarajan[1†], Andrew P Hederman[2], Carly A Bobak[3], Joshua A Weiner[2], Wendy Wieland-Alter[4], Jiwon Lee[2], Evan M Bloch[5], Aaron AR Tobian[5], Andrew D Redd[6,7], Joel N Blankson[6], Dana Wolf[8], Tessa Goetghebuer[9,10], Arnaud Marchant[9], Ruth I Connor[4], Peter F Wright[4], Margaret E Ackerman[1,2,3]*

[1]Department of Microbiology and Immunology, Geisel School of Medicine at Dartmouth, Dartmouth College, Hanover, United States; [2]Thayer School of Engineering, Dartmouth College, Hanover, United States; [3]Biomedical Data Science, Dartmouth College, Hanover, United States; [4]Department of Pediatrics, Geisel School of Medicine at Dartmouth, Dartmouth-Hitchcock Medical Center, Lebanon, United States; [5]Department of Pathology, Johns Hopkins School of Medicine, Baltimore, United States; [6]Department of Medicine, Division of Infectious Diseases, Johns Hopkins School of Medicine, Baltimore, United States; [7]Division of Intramural Research, National Institute of Allergy and Infectious Diseases, National Institutes of Health, Bethesda, United States; [8]Hadassah University Medical Center, Jerusalem, Israel; [9]Institute for Medical Immunology, Université libre de Bruxelles, Charleroi, Belgium; [10]Pediatric Department, CHU St Pierre, Brussels, Belgium

*For correspondence:
Margaret.E.Ackerman@
dartmouth.edu

[†]These authors contributed
equally to this work

Competing interest: The authors
declare that no competing
interests exist.

Reviewing Editor: Tomohiro
Kurosaki, Osaka University, Japan

**Abstract** Preexisting antibodies to endemic coronaviruses (CoV) that cross-react with SARS-CoV-2 have the potential to influence the antibody response to COVID-19 vaccination and infection for better or worse. In this observational study of mucosal and systemic humoral immunity in acutely infected, convalescent, and vaccinated subjects, we tested for cross-reactivity against endemic CoV spike (S) protein at subdomain resolution. Elevated responses, particularly to the β-CoV OC43, were observed in all natural infection cohorts tested and were correlated with the response to SARS-CoV-2. The kinetics of this response and isotypes involved suggest that infection boosts preexisting antibody lineages raised against prior endemic CoV exposure that cross-react. While further research is needed to discern whether this recalled response is desirable or detrimental, the boosted antibodies principally targeted the better-conserved S2 subdomain of the viral spike and were not associated with neutralization activity. In contrast, vaccination with a stabilized spike mRNA vaccine did not robustly boost cross-reactive antibodies, suggesting differing antigenicity and immunogenicity. In sum, this study provides evidence that antibodies targeting endemic CoV are robustly boosted in response to SARS-CoV-2 infection but not to vaccination with stabilized S, and that depending on conformation or other factors, the S2 subdomain of the spike protein triggers a rapidly recalled, IgG-dominated response that lacks neutralization activity.

## Editor's evaluation

This study is aimed to determine whether infection or vaccination affects activation of preexisting memory B cells. The data are clear and have implications for further development of a new generation of vaccines.

## Introduction

The ongoing pandemic of SARS-CoV-2 represents the third time in just two decades that a novel coronavirus (CoV) with significant morbidity and mortality has begun to circulate among humans (*Peeri et al., 2020*). Given the alarming frequency of these occurrences, the COVID-19 pandemic serves as a call to action to safeguard against continued emergence of novel human CoV. Unlike in the earlier outbreaks of SARS-CoV and MERS-CoV, however, SARS-CoV-2 has proven to be highly transmissible, infecting a substantial portion of the global population – a conservative estimate of the prevalence of infection based on only confirmed cases corresponds to over 2% (*COVID, 2021*), while estimates of the total figure in the United States exceed 16% of the population (*Reese et al., 2021*). The extent to which the immune responses generated by these and other endemic CoV exposures might serve as an effective deterrent against the emergence of novel CoV strains remains an open question.

Insights into the effect of exposure to endemic CoVs in the years leading up to the COVID-19 pandemic could suggest whether preexisting antibodies are likely to contribute beneficially or detrimentally to outcomes of infection by a novel strain. Under the hypothesis of original antigenic sin, preexisting humoral memory imprinted from prior exposure to related antigens partially predestines the antibody repertoire to focus on the epitopes of a new threat that closely resemble those for which there is an existing solution. Because neutralizing epitopes are subject to additional selective pressure, this recall and re-diversification of existing antibody lineages may hinder the immune system's ability to generate effective neutralizing antibodies (*Kim et al., 2009*). This phenomenon has been most convincingly established in the context of influenza virus infection, with support from both well-controlled animal model experiments (*Kim et al., 2009*; *Nachbagauer et al., 2017*) and observational studies of human natural infection histories (*Francis et al., 1947*; *Fonville et al., 2014*; *Kucharski et al., 2015*; *Gostic et al., 2016*; *Lee et al., 2019*). Similar observations have been made in the context of diverse dengue virus serotypes (*Midgley et al., 2011*). These and other studies have shown that despite being a key hallmark of effective long-term immune defense, anamnestic responses are not without potential downsides.

Antibody-dependent enhancement (ADE), a phenomenon by which cross-reactive antibodies induced by a prior exposure to a related pathogen promote infection of cell types bearing antibody Fc receptors and potentially elevate morbidity and mortality (reviewed in *Bournazos et al., 2020*; *Taylor et al., 2015*), suggests further potential advantages of a humoral blank slate. Though observed in vitro (*Wu et al., 2020*; *Li et al., 2021*), there has not been abundant evidence for the biological relevance of classical ADE in the context of SARS-CoV-2 in vivo (*Li et al., 2021*; *Wang et al., 2016*). Nonetheless, the role of non-neutralizing antibodies resulting from cross-strain challenges in promoting ADE in cases of dengue fever (*Katzelnick et al., 2017*) has motivated concern about COVID-19 enhancement (*Lee et al., 2020*; *Fierz and Walz, 2020*; *Arvin et al., 2020*).

The extent to which preexisting responses to prior endemic CoV exposures may influence responses to SARS-CoV-2 infection and vaccination is not yet clear, but has been suggested from studies relating recent endemic CoV infection with reduced severity of COVID-19 (*Sagar et al., 2021*; *Aran et al., 2020*). Numerous studies have observed elevated responses to endemic CoV following SARS-CoV-2 infection (*Guo et al., 2021*; *Morgenlander et al., 2021*; *Wang et al., 2021*; *Kaplonek et al., 2021*; *Ortega et al., 2021*), and more recent work has shown that the magnitude of this 'back-boosting' effect is inversely correlated with the induction of IgG and IgM against SARS-CoV-2 spike (S) protein (*Aydillo et al., 2021*). Given differential degrees of homology between the receptor-binding domain (RBD) in S1 that is the target of the majority of neutralizing antibodies, and the better-conserved S2 domain that may be the target of antibodies with diverse effector functions but which are rarely neutralizing, the original antigenic sin hypothesis suggests that lower titers of neutralizing antibodies against SARS-CoV-2 may result from boosting of preexisting cross-reactive lineages. Here, this apparent boosting effect is evaluated in diverse cohorts with the goal of beginning to understand its implications.

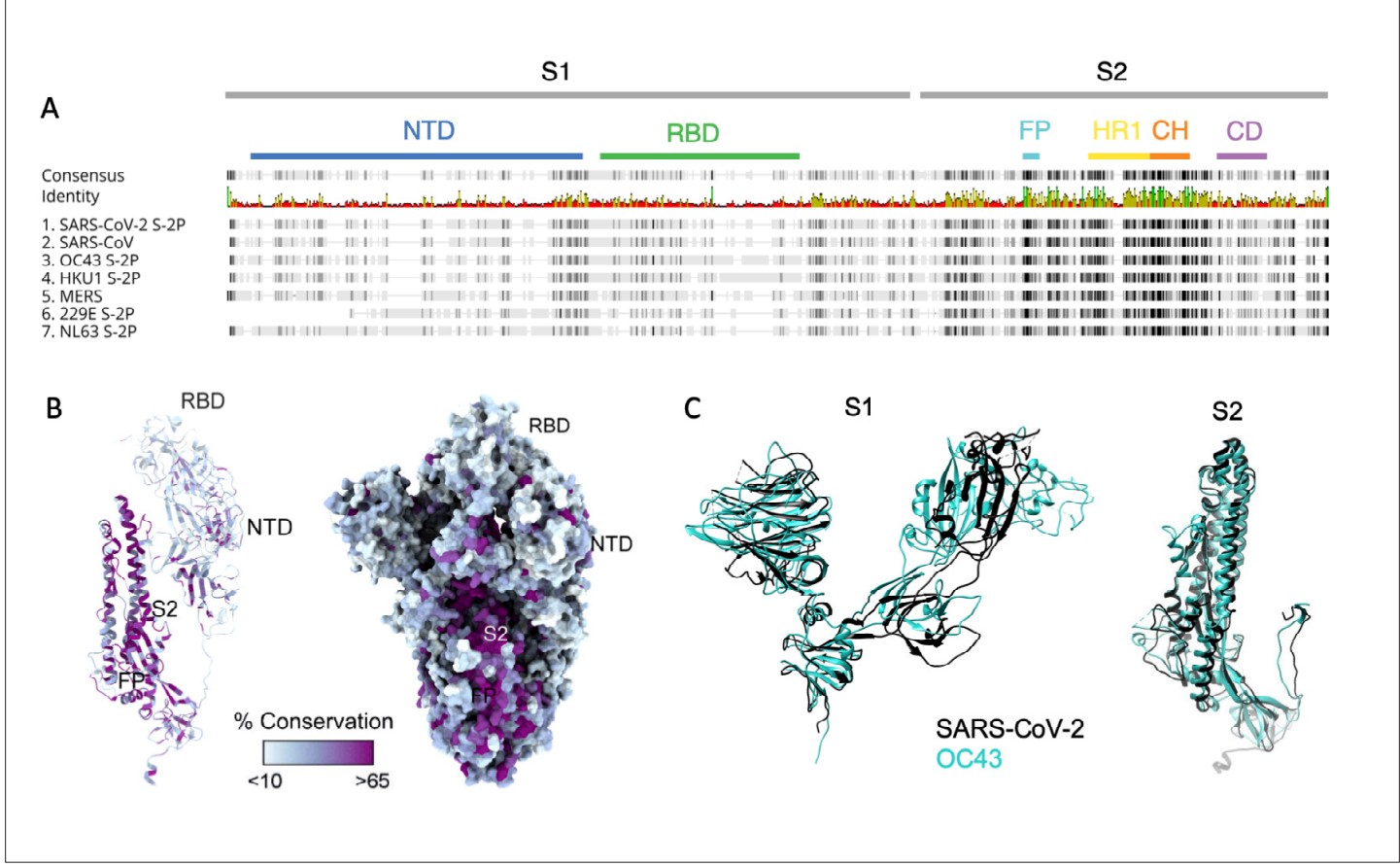

**Figure 1.** Sequence and structural differences between the spike ectodomain of SARS-CoV-2 and endemic strains. (**A**) Sequence alignment of SARS-CoV-2 spike protein to other human coronaviruses. Consensus identity is shown on a scale from red (least conserved) to green (most conserved). Color bars are used to indicate different regions of the spike protein: N-terminal domain (NTD, blue), receptor-binding domain (RBD, green) in the S1 domain and fusion peptide (FP, cyan), heptad repeat 1 (HR1, yellow), central helix (CH, orange), and connector domain (CD, purple) in the S2 domain. (**B**) Structural model of the spike protein monomer colored by percent sequence conservation across deposited coronaviridae sequences shown as a ribbon model for one protomer (left) and spacefill for the spike trimer (right). (**C**) Superimposed structural model of the spike protein S1 (left) and S2 (right) domains for SARS-CoV-2 (black) and OC43 (teal).

The online version of this article includes the following figure supplement(s) for figure 1:

**Figure supplement 1.** Structural differences between the S1 domain of SARS-CoV-2 and endemic strains.

## Results

### Structural analysis of SARS-CoV-2

Sequence conservation across the human CoV spike protein is not uniformly distributed: the N-terminal domain (NTD) of SARS-CoV-2 that contains the RBD responsible for interacting with human angiotensin-converting enzyme 2 (ACE2) and initiating viral entry has lower homology to the corresponding subdomains of other CoVs than do the S2 subdomains where the fusion peptide, heptad repeat, and central helix required for fusion are located (*Figure 1A*). In contrast to the S2 domain, structural comparison of various subdomains of the SARS-CoV-2 S protein shows that both the NTD and RBD of the spike protein are poorly conserved across CoV (*Figure 1B*). Superimposition of the SARS CoV-2 S1 and S2 domains with the most well-conserved widely circulating endemic human CoV, OC43, show high structural conservation in S2 and the NTD, and a complete lack of homology in the RBD that is consistent with the differing entry receptors used by these β-CoV (*Figure 1C*, *Figure 1—figure supplement 1*). Based on both structural and sequence homology, it stands to reason that preexisting antibodies raised against endemic human CoV are more likely to target the better-conserved regions of SARS-CoV-2 such as S2, and less likely to recognize the RBD (*Yuan et al., 2020*; *Wrapp et al., 2020*).

## Subject cohorts

A diversity of cohorts were evaluated in this observational study (*Table 1*). Subjects naturally infected with SARS-CoV-2 comprised a small cohort of convalescent subjects for which serum and mucosal samples were available, a larger cohort of convalescent plasma donors, acutely infected individuals, pregnant women infected in their third trimester, and a small set of subjects for whom pre- and post-infection samples were available. Subjects vaccinated against SARS-CoV-2 with mRNA included cohorts of healthy adults and pregnant women vaccinated in their third trimester. Samples were analyzed alongside samples from naïve (n = 15) and commercial (n = 38) negative controls.

## Elevated responses to endemic CoV in serum, nasal wash, and stool among SARS-CoV-2-infected subjects

The magnitude and specificity of IgM, IgA, and IgG responses were determined across a panel of SARS-CoV-2 and endemic CoV antigens in two previously described cohorts of convalescent subjects (*Butler, 2020*; *Klein et al., 2020*; *Table 1*). In the smaller (Dartmouth-Hitchcock Medical Center [DHMC], n = 26) cohort, mucosal samples were available, and responses in nasal wash and stool were also defined. Relative to SARS-CoV-2-naïve controls, elevated serum IgA and IgG but not IgM responses to whole S of diverse endemic CoV were frequently observed (*Figure 2*). Among endemic CoV, elevated levels of OC43-specific responses in serum were most pronounced, but were largely restricted to the S2 domain and whole unstabilized S. Similarly, elevated IgG responses to other endemic CoV were observed in serum from convalescent subjects to the spike proteins of endemic CoV, but not to the S1 domain alone. Indeed, elevated serum responses specific to the S1 domain of these CoV were not observed for any antibody isotype or subclass (*Figure 2B*, *Figure 2—figure supplement 1*).

These observations were validated in a larger (Johns Hopkins Medical Institutions [JHMI], n = 126) cohort of convalescent plasma donors: elevated levels of IgG and IgA, but not IgM, to diverse endemic CoV were observed in plasma (*Figure 2B*, *Figure 2—figure supplement 2*). Elevated responses were most pronounced for β-CoV (OC43 and HKU1) but were present for α-CoV as well. The absence of elevated IgM responses among convalescent subjects is consistent with recall of class-switched antibodies as opposed to the de novo elicitation of cross-reactive antibodies. Additionally, differential reactivity profiles were observed across three different OC43 antigens: proline-stabilized OC43 S (S-2P), OC43 S, and OC43 S2. Whereas elevated responses to OC43 S-2P among convalescent subjects in either cohort were not observed compared to naïve controls, responses to both OC43 S and OC43 S2 (available only for the smaller cohort) were observed for IgG, but not IgM (*Figure 2— figure supplement 3*).

This effect was not limited to serum; elevation of endemic CoV antibody responses was also observed in nasopharyngeal wash samples (*Figure 2A and B*, *Figure 2—figure supplement 1*) and stool (*Figure 2—figure supplement 1*). Again, these elevated responses were restricted to IgG and/ or IgA isotypes, consistent with cross-reactive antibodies being boosted in a recall response during SARS-CoV-2 infection, and were observed for whole S or the S2 domain, but not S1.

## OC43-specific antibody responses in acute infection

To better explore the kinetics of these responses, antibodies were measured in serum collected from a cohort of acutely infected subjects (n = 10, *Table 1*) 2 weeks after a positive PCR diagnosis afforded by twice-weekly surveillance. At this early timepoint, subjects appeared to have robustly elevated levels of CoV-2 unstabilized S-specific IgG compared to naïve controls (*Figure 3A*). Again, these responses were observed for whole unstabilized S and the S2 domain, but not for the prefusion conformation-stabilized S-2P form of OC43 spike. In contrast, only a minor increase in IgM recognizing unstabilized OC43 S was observed (less than twofold) (*Figure 3A*). Though cross-sectional in nature and reliant on a small number of subjects, these isotype profiles in acute infection nonetheless further suggest a recalled rather than novel response against the relevant cross-reactive epitopes in S2 and unstabilized spike, presumably in the postfusion conformation.

In an effort to resolve whether elevated responses to endemic β-CoV were driven by boosting of preexisting cross-reactive antibody lineages, a small cohort of subjects (n = 3) for which pre- and post-SARS-CoV-2 infection serum samples were available was analyzed (*Table 1*). The median increase in OC43 S2-specific IgA and IgG were 3.7- and 20-fold, respectively, between the pre- and post-infection

**Table 1.** Cohort characteristics.

NA: not applicable or available; IQR: interquartile range; DHMC: Dartmouth-Hitchcock Medical Center; JHMI: Johns Hopkins Medical Institutions. Partially reproduced from *Natarajan et al., 2021*.

| Characteristic | DHMC naive | DHMC convalescent | JHMI convalescent | Acute | Pre- and post-infection | Pregnant infected | Pregnant vaccinated | Vaccinated | Commercial controls |
|---|---|---|---|---|---|---|---|---|---|
| | n = 15 | n = 26 | n = 126 | n = 10 | n = 3 | n = 38 | n = 50 | n = 37 | n = 38 |
| Median age (IQR), years | 34 (28–52) | 58 (18–77) | 42 (29–53) | 27 (20–30) | 62 (61–64) | 31 (27–35) | 32 (29–35) | NA | 39 (28–50) |
| Sex (n, %) | | | | | | | | | |
| Female | 8 (53%) | 13 (50%) | 58 (46%) | 3 (30%) | 2 (67%) | 38 (100%) | 50 (100%) | 17 (46%) | 22 (58%) |
| Male | 7 (47%) | 13 (50%) | 68 (54%) | 7 (70%) | 1 (33%) | 0 (0%) | 0 (0%) | 20 (54%) | 16 (42%) |
| Hospitalized (severity) | | | | | | | | | |
| No | NA | 20 (77%) | 114 (90.5%) | 10 (100%) | 3 (100%) | NA | NA | NA | NA |
| Yes | | 6 (23%) | 12 (9.5%) | 0 (0%) | 0 (0%) | | | | |
| Median days since PCR+ or symptom onset (IQR) | NA | 42.5 (19–154) | 43 (38–48) | 11 (9–14) | 12 (11–14) | 49 (22–78) | NA | NA | NA |
| Median days since second vaccine dose (IQR) | NA | NA | NA | NA | NA | NA | 20 (12–29) | 8 (7–11) | NA |
| Location | US | US | US | US | US | Belgium | Israel | US | US |
| IRB | DHMC | DHMC | JHMI | DHMC | DHMC | CHU St. Pierre | Hadassah Medical Center | JHMI | BioIVT clinical sites |

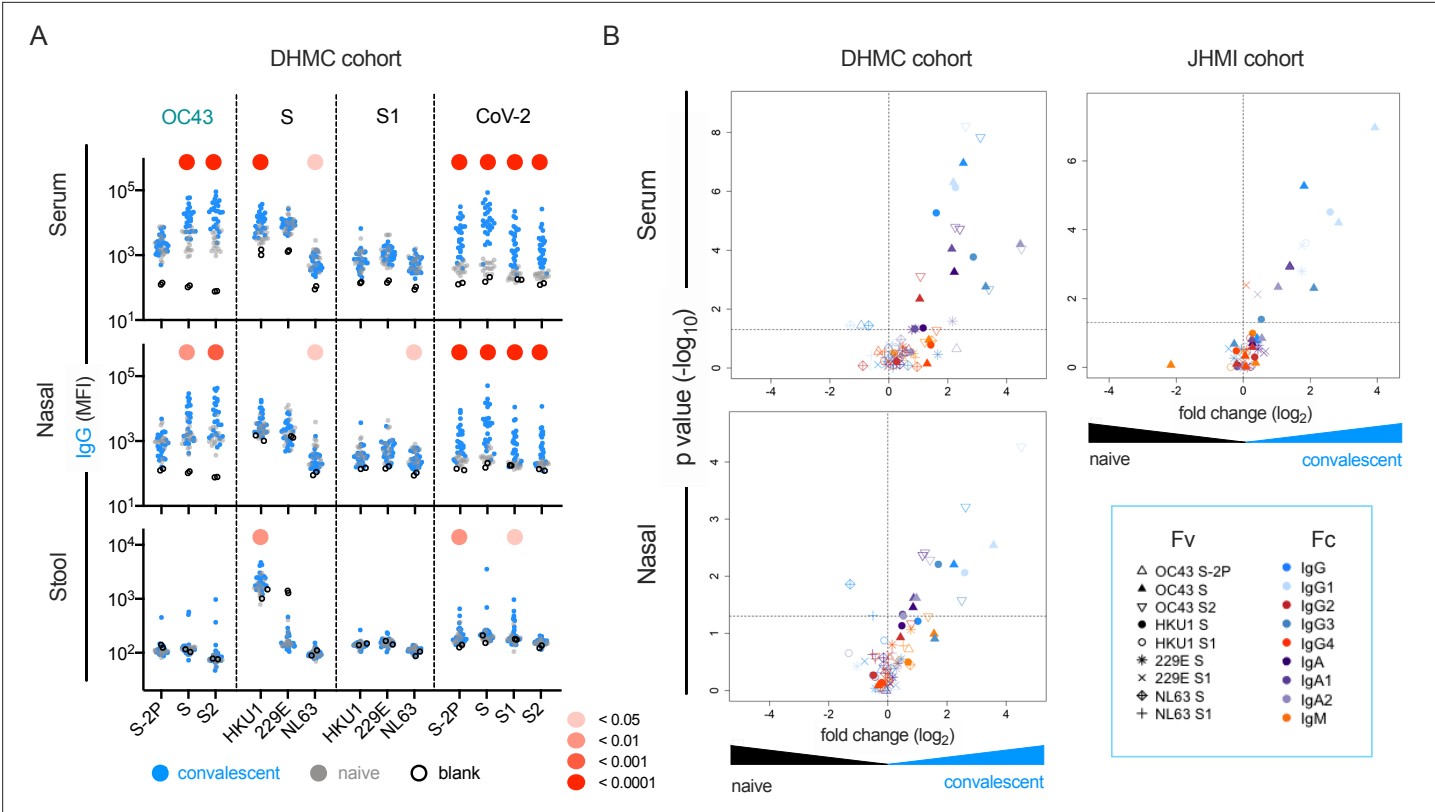

**Figure 2.** this SARS-CoV-2 infection is associated with elevated IgG and IgA responses to endemic CoV. (**A**) IgG responses in serum (top), nasal wash (middle), and stool (bottom) across antigens from CoV-2, OC43, and other endemic CoV S, and S1 proteins in the Dartmouth-Hitchcock Medical Center (DHMC) convalescent cohort. Samples from naïve subjects are indicated in gray, SARS-CoV-2 convalescents at 1 month post infection in blue, and buffer blanks in hollow circles. (**B**) Volcano plot of fold change and significance (unpaired *t*-test) of differences between antibody responses observed in convalescent subjects of the DHMC (left) and Johns Hopkins Medical Institutions (JHMI) (right) cohorts 1 month post infection and naïve subjects in serum (top) and nasal wash (bottom). Each symbol represents an antibody response feature, with Fc domain characteristics represented by color and Fv antigen specificity indicated by shape. Dotted horizontal line illustrates p=0.05. Statistical significance was defined by Mann–Whitney *U*-test.

The online version of this article includes the following figure supplement(s) for figure 2:

**Figure supplement 1.** IgA and IgM responses in the Dartmouth-Hitchcock Medical Center (DHMC) cohort.

**Figure supplement 2.** Antibody responses in the Johns Hopkins Medical Institutions (JHMI) cohort.

**Figure supplement 3.** Elevated IgG but not IgM responses to endemic CoV in convalescent cohorts.

---

timepoints (***Figure 3B***). In contrast, the median change in OC43 S2-specific IgM was within twofold, providing further support that the elevated responses to endemic β-CoV are due to boosting of preexisting cross-reactive antibodies originally raised against homologous epitopes found in endemic CoV. Like in other convalescent subject cohorts, this boosting effect was observed in response to OC43 S and S2, but not the prefusion conformation-stabilized S-2P form of OC43, which showed a median change less than 1.1-fold across isotypes. Boosting of IgG responses specific to the full-length spike of another endemic β-CoV, HKU1, was also observed. IgG and IgA but not IgM responses to whole unstabilized S, but not the S1 domain, of HKU1 were elevated following SARS-CoV-2 infection (***Figure 3B***, ***Figure 3—figure supplement 1***). Collectively, these results further suggest that recalled cross-reactive antibodies may be more likely to recognize both the better-conserved S2 domain and post- rather than prefusion conformations of spike.

## Correlations between antibody responses to OC43 and SARS-CoV-2

We next examined correlative relationships between SARS-CoV-2- and OC43-specific IgG responses in convalescent cohorts. Whereas the magnitude of IgG responses to CoV-2 S2 were well correlated to those binding unstabilized OC43 S ($R_P = 0.61$) and its S2 domain ($R_P = 0.45$), they were less well correlated to responses to stabilized OC43 S-2P ($R_P = 0.29$) (***Figure 4A***). Expanding this analysis to

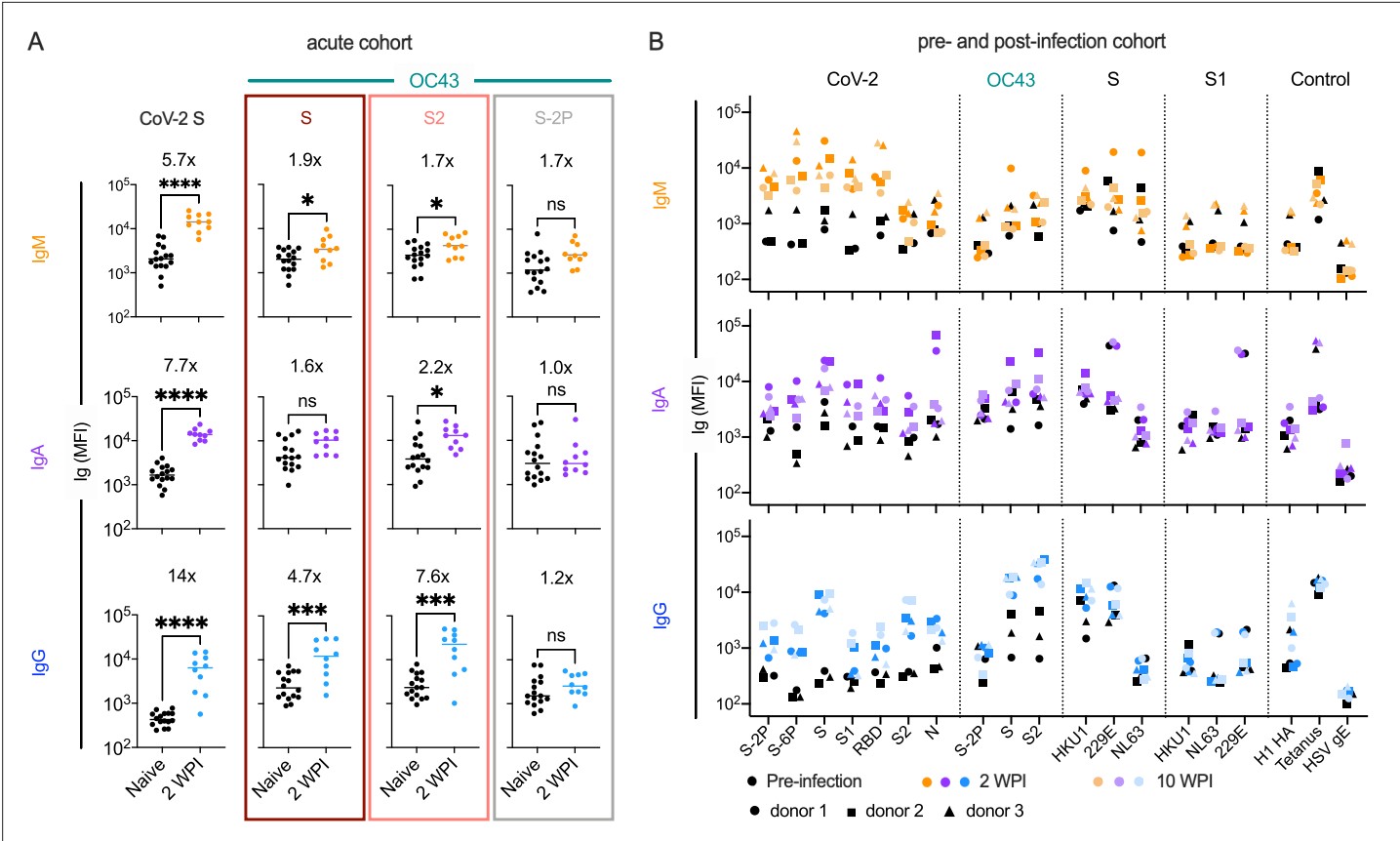

**Figure 3.** Response kinetics and isotype profiles in acute and pre- and post-infection samples suggest recall of preexisting, cross-reactive antibodies. (**A**) OC43 and CoV-2-specific IgM (orange), IgA (purple), and IgG (blue) responses in 10 acutely infected subjects (color) 2 weeks post infection (WPI) as compared to naïve subjects (black). Statistical significance was assessed by unpaired *t*-test (*p<0.05, ***p<0.0005, ****p<0.00005). Fold change in means between groups is presented in inset. (**B**) IgM (orange), IgA (purple), and IgG (blue) responses across CoV-2, other CoV, and control antigens in three subjects (indicated by shape) pre- (black) and post- (color) SARS-CoV-2 infection. Serum samples were taken 2 and 10 WPI.

The online version of this article includes the following figure supplement(s) for figure 3:

**Figure supplement 1.** Elevated responses to endemic CoV in the pre- and post-infection cohort.

include additional CoV-2 specificities and other isotypes showed a hierarchy of correlative relationships (***Figure 4B***, ***Figure 4—figure supplement 1***). CoV-2-specific responses were better correlated to OC43 responses specific for unstabilized rather than stabilized spike. IgG responses showed stronger relationships than did IgA, which were in turn stronger than IgM. Consistent with the lack of elevated IgM to endemic CoV, these measures only rarely showed a statistically significant relationship with CoV-2-specific IgM responses. Lastly, among CoV-2 antigens tested, correlations with OC43 responses were strongest for the S2 domain and whole spike (S-2P), and weaker or absent for S1 and the RBD. Collectively, these correlative relationships are consistent with recall of class-switched antibodies recognizing shared epitopes in the S2 domain from prior endemic CoV exposure induced by SARS-CoV-2 infection.

## Direct evidence of molecular cross-reactivity of SARS-CoV-2 and OC43-specific antibodies

To date, antibody cross-reactivity has been inferred from indirect evidence in the form of boosted responses to endemic CoV (***Guo et al., 2021***; ***Morgenlander et al., 2021***; ***Wang et al., 2021***; ***Kaplonek et al., 2021***; ***Ortega et al., 2021***), and more conclusively observed for select monoclonal antibodies that have been cloned and cross-tested (***Sakharkar et al., 2021***; ***Dugan et al., 2021***). To better generalize the more definitive monoclonal studies, we sought to directly define the cross-reactivity of polyclonal antibodies raised following SARS-CoV-2 infection. Antibodies specific to stabilized

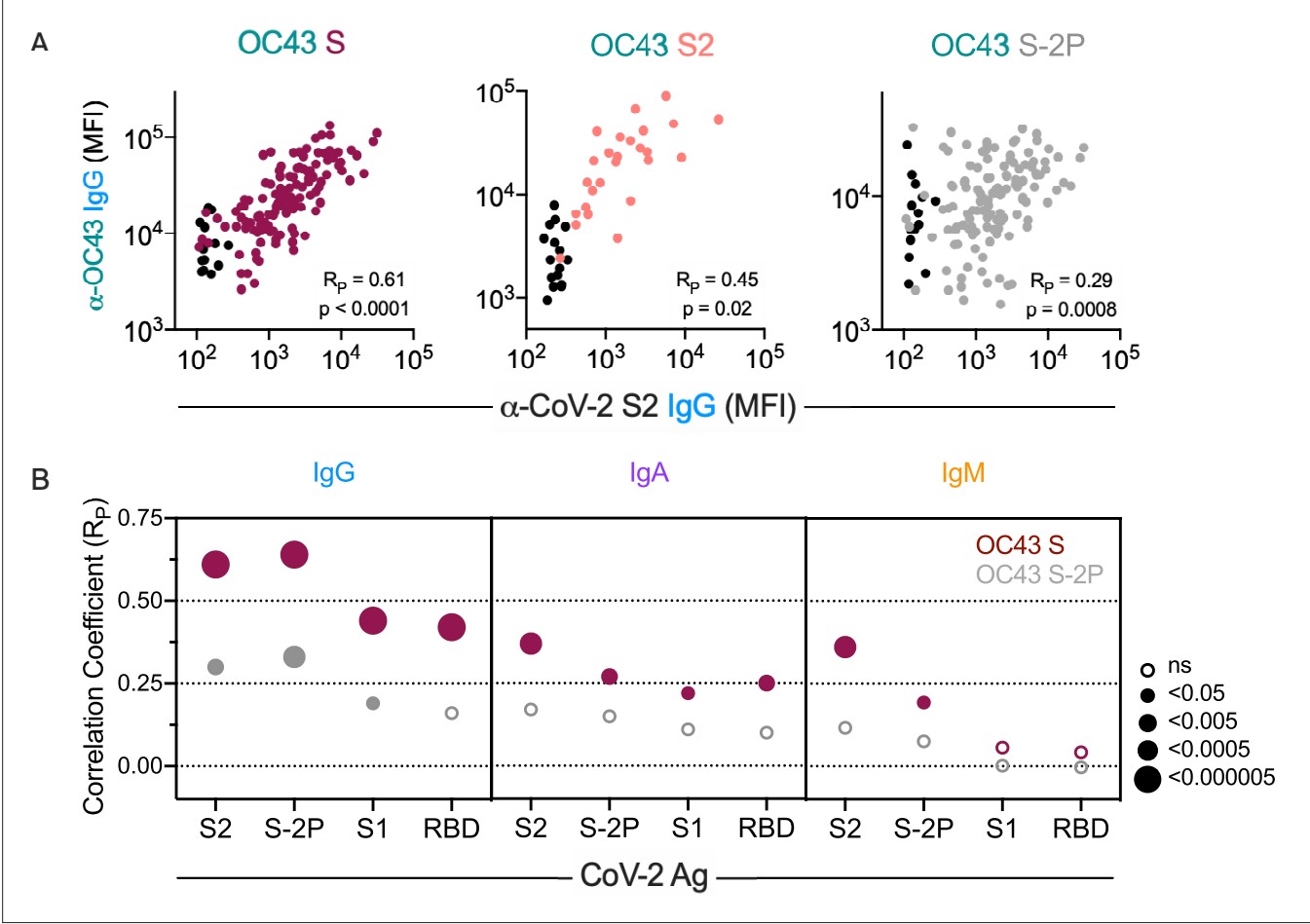

**Figure 4.** Correlative relationships between CoV-2 and OC43-specific antibody features. (**A**) Scatterplots of IgG responses specific to OC43 S, OC43 S2, and OC43 S-2P versus CoV-2 S2. Naïve subjects were excluded from calculations of correlative relationships. (**B**) Correlations ($R_P$) between IgG, IgA, and IgM specific to different stabilized SARS-CoV-2 spike and its subdomains with responses to OC43 S (maroon) and OC43 S-2P. Size and fill of symbols indicate statistical significance. Responses and relationships for naïve subjects are shown in black and convalescent donors shown in maroon (OC43 S), salmon (OC43 S2), and gray (OC43 S-2P).

The online version of this article includes the following figure supplement(s) for figure 4:

**Figure supplement 1.** Correlative relationships between SARS-CoV-2- and OC43-specific antibody responses in the Johns Hopkins Medical Institutions (JHMI) cohort by isotype.

CoV-2 S-2P, RBD, and S2 and unstabilized OC43 S were selectively purified from serum samples from 30 SARS-CoV-2-infected subjects with a range of disease severity and humoral response profiles. Unfractionated and antigen-specific antibodies eluted from affinity purification matrices presenting various epitopes of the CoV-2 spike protein and OC43 were then characterized to determine their cross-reactivity and isotype profiles. Successful affinity purification was confirmed by comparison of antigen-specific binding signal relative to total Ig levels for each isotype (***Figure 5A***, ***Figure 5—figure supplements 1–12***). While relative binding signal was elevated for each targeted antigen, it was not observed for control antigens such as influenza hemagglutinin (HA) (***Figure 5A***) or tetanus toxoid (***Figure 5—figure supplements 1–12***).

A number of interesting differences in cross-reactivity for antibodies with differing isotypes and antigen-specificities were apparent. For example, OC43 S-specific fractions showed elevated recognition of CoV-2 S2 and S-2P, but not RBD (***Figure 5A***), demonstrating molecular cross-reactivity as a general feature of polyclonal IgG responses in convalescent subjects. Additionally, IgG in the CoV-2 S2-specific fractions cross-reacted robustly with OC43 S, but no cross-reactivity in the IgM fraction was observed, despite a robust IgM response to CoV-2 S2 (***Figure 5B***), suggesting that while SARS-CoV-2

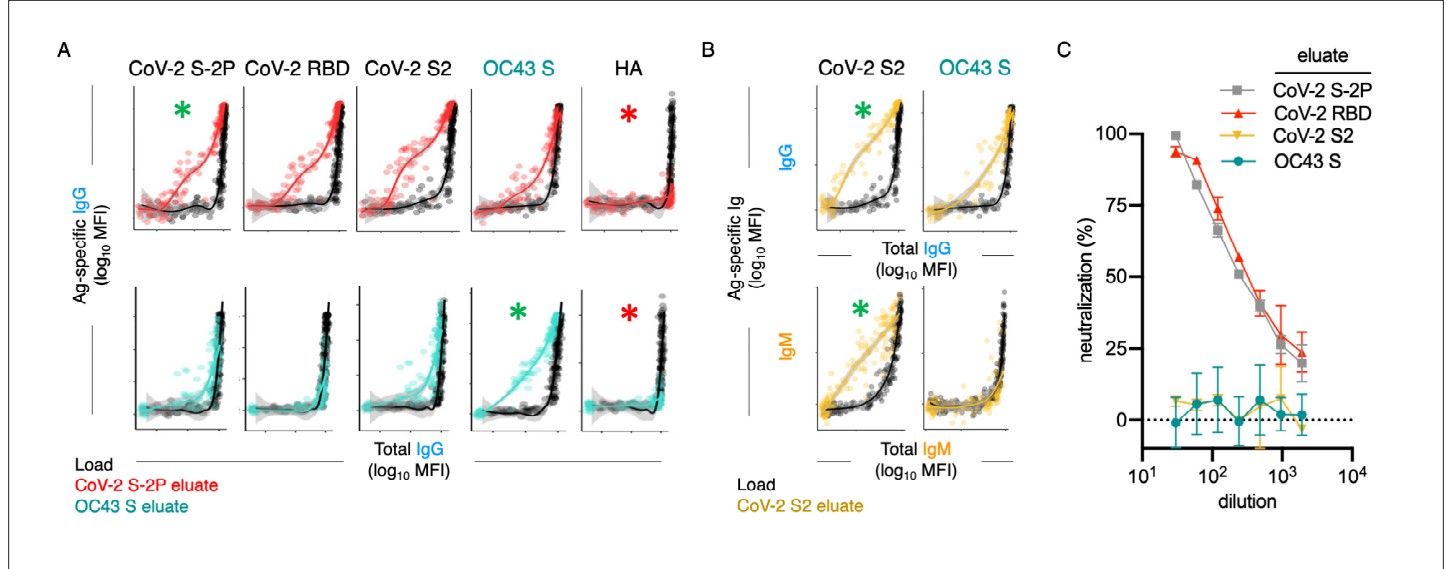

**Figure 5.** Poor neutralization activity of cross-reactive and S2-specific antibodies established by affinity purification. (**A**) Antigen-binding profiles of IgG in unfractionated serum (load, black) and affinity-purified CoV-2 S-2P- (red, top) and OC43 S- (teal, bottom) fractions (eluate) from 30 SARS-CoV-2 convalescent subjects. Reactivity to CoV-2 S-2P, CoV-2 RBD, CoV-2 S2, OC43 S, and a control antigen (influenza hemagglutinin [HA]) are reported. (**B**) Antigen-binding profiles of IgG (top) and IgM (bottom) in unfractionated serum (load, black) and affinity-purified CoV-2 S2-specific (yellow) eluate. Reactivity to CoV-2 S2 (left) and OC43 S (right) is shown. For (**A**, **B**), responses to the matched antigen (positive control) used in purification are indicated by green asterisks and to HA (negative control) antigen with red asterisks. Smoothed curves and 95% confidence intervals are shown for both eluate and load fractions. (**C**) Neutralization activity of pooled elution fractions of antibodies affinity-purified against CoV-2 S-2P (gray square), CoV-2 RBD (red triangle), CoV-2 S2 (yellow triangle), and OC43 S (teal circle). Error bars depict standard error of the mean across assay duplicates.

The online version of this article includes the following figure supplement(s) for figure 5:

**Figure supplement 1.** Cross-reactivity of SARS-CoV-2 S-2P-specific IgG.

**Figure supplement 2.** Cross-reactivity of SARS-CoV-2 receptor-binding domain (RBD)-specific IgG.

**Figure supplement 3.** Cross-reactivity of SARS-CoV-2 S2-specific IgG.

**Figure supplement 4.** Cross-reactivity of OC43 S-specific IgG.

**Figure supplement 5.** Cross-reactivity of SARS-CoV-2 S-2P-specific IgA.

**Figure supplement 6.** Cross-reactivity of SARS-CoV-2 receptor-binding domain (RBD)-specific IgA.

**Figure supplement 7.** Cross-reactivity of SARS-CoV-2 S2-specific IgA.

**Figure supplement 8.** Cross-reactivity of OC43 S-specific IgA.

**Figure supplement 9.** Cross-reactivity of SARS-CoV-2 S-2P-specific IgM.

**Figure supplement 10.** Cross-reactivity of SARS-CoV-2 receptor-binding domain (RBD)-specific IgM.

**Figure supplement 11.** Cross-reactivity of SARS-CoV-2 S2-specific IgM.

**Figure supplement 12.** Cross-reactivity of OC43 S-specific IgM.

infection elicits IgM responses to the S2 domain, these presumed de novo responses are not cross-reactive to OC43.

To begin to define the functional significance of these cross-reactive antibodies, eluted fractions were pooled and tested for neutralization potency. Whereas purified CoV-2 RBD and S-2P-specific fractions induced robust neutralization, neither purified OC43 S-specific nor SARS-CoV-2 S2 domain-specific serum antibodies had detectable neutralization activity (***Figure 5C***).

To quantify the extent of cross-reactivity across antigens and isotypes, the unfractionated and affinity-purified fractions from each subject were titrated, and enrichment of antigen-specific antibodies was computed by calculating the difference between the area under the curve (AUC) for the best-fit lines calculated from data points across subjects for each antibody specificity and each isotype in eluates versus unfractionated samples (***Figure 6***).

IgM, IgA, and IgG antibodies purified based on binding to SARS-CoV-2 RBD cross-reacted with proline-stabilized spike (S-2P and S-6P) and, to a lesser extent, unstabilized S, but showed no evidence

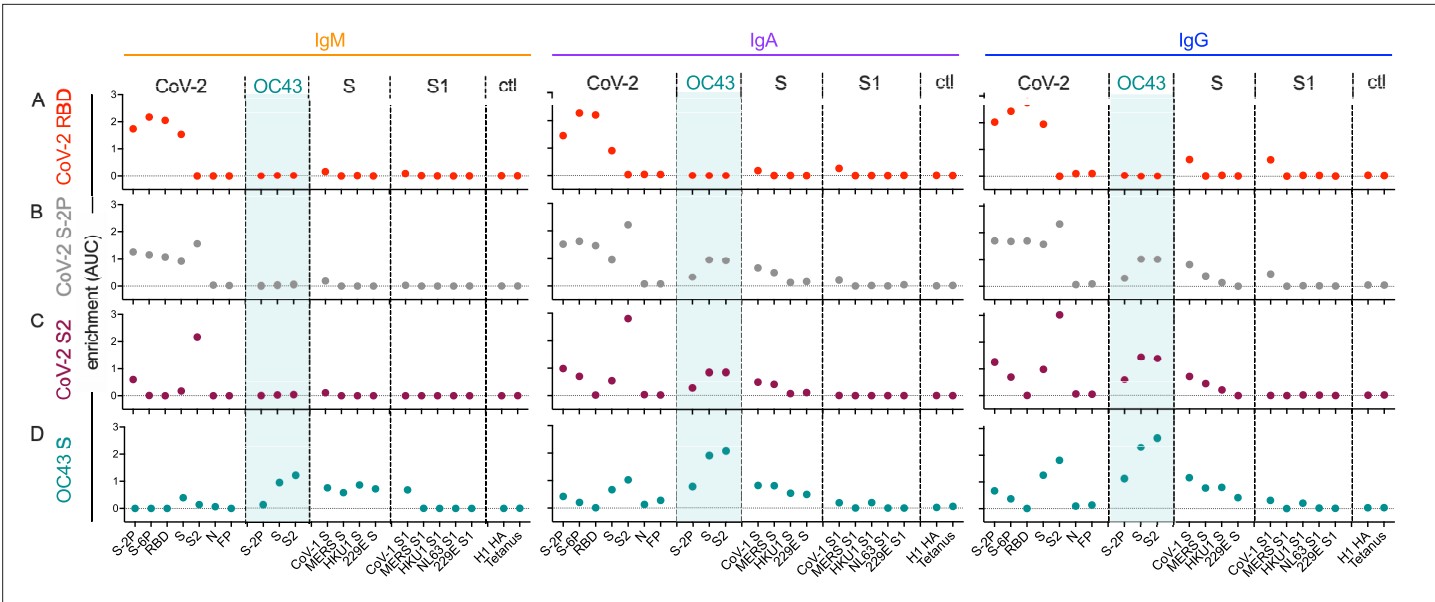

**Figure 6.** Molecular cross-reactivity profiles of OC43 and SARS-CoV-specific antibody fractions across antigen specificities and Ig isotypes. The degree of enrichment (area under the curve, AUC) of affinity-purified SARS-CoV-2 receptor-binding domain (RBD-) (**A**), S-2P- (**B**), S2 domain- (**C**), or OC43 S- (**D**) specific antibodies of IgM (left), IgA (center), and IgG (right) isotypes across diverse CoV and control (ctl) proteins. Dotted line indicates no enrichment.

of cross-reactivity to SARS-CoV-2 S2 or to endemic CoV (**Figure 6A**). Because the overwhelming majority of neutralizing antibodies target the RBD, these results suggest that RBD-specific antibody responses raised by natural infection are unlikely to exhibit exceptionally broad CoV neutralization activity. Indeed, SARS-CoV-2 RBD-specific antibodies showed low, albeit detectable, cross-reactivity toward even SARS-CoV S and S1.

Similarly, across all isotypes, antibody pools purified based on binding to stabilized SARS-CoV-2 S-2P showed the expected cross-reactivity to SARS-CoV-2 RBD, S2, unstabilized S, and stabilized S with six proline substitutions (S-6P) (**Figure 6B**). However, while both IgA and IgG components of SARS-CoV-2 S-2P-specific serum antibody fractions showed binding to OC43, cross-reactive IgM was not evident. Similarly, IgA and IgG, but not IgM, specific for SARS-CoV-2 S2 bound as expected to SARS-CoV-2 proteins as well as OC43, SARS-CoV, MERS, and other endemic CoV S2 domains (**Figure 6C**).

Lastly, OC43 S-specific antibodies were similarly purified and profiled for cross-reactivity. Despite the presumption that subjects had not experienced OC43 infection recently, IgM, IgA, and IgG to OC43 S were all successfully enriched (**Figure 6D**). IgA and IgM fractions showed cross-reactivity to a broad array of endemic CoV S but not S1 proteins. Likewise, OC43-specific IgA and IgG antibodies showed cross-reactivity to SARS-CoV-2 proteins containing the S2 domain. In contrast, OC43 S-specific IgM antibodies showed elevated recognition of diverse pandemic, pathogenic, and endemic S proteins in unstabilized forms, but a lack of recognition of stabilized prefusion conformations of CoV-2 spike, or the S1 domains of most other CoV tested. Given binding to whole S but not the S1 domain, this recognition is presumably principally driven via recognition of S2.

## Immunization with stabilized spike changes cross-reactivity profiles

Based on the differential ability of affinity-purified antibodies to recognize stabilized and unstabilized forms of CoV spike proteins, we next sought to determine whether responses to natural infection differ from those that result from mRNA vaccination with stabilized SARS-CoV-2 spike (S-2P). To investigate this possibility, antibody responses were analyzed in two cohorts of pregnant women (**Table 1**) who were either infected (n = 38) or vaccinated (n = 50) during their third trimester, as well as a validation cohort of healthy vaccinated subjects (n = 37). Fascinatingly, despite inducing considerably greater levels of SARS-CoV-2-specific antibodies, immunization with stabilized spike (S-2P) failed to result in elevated IgG responses to endemic CoV among pregnant women (**Figure 7**). Similarly,

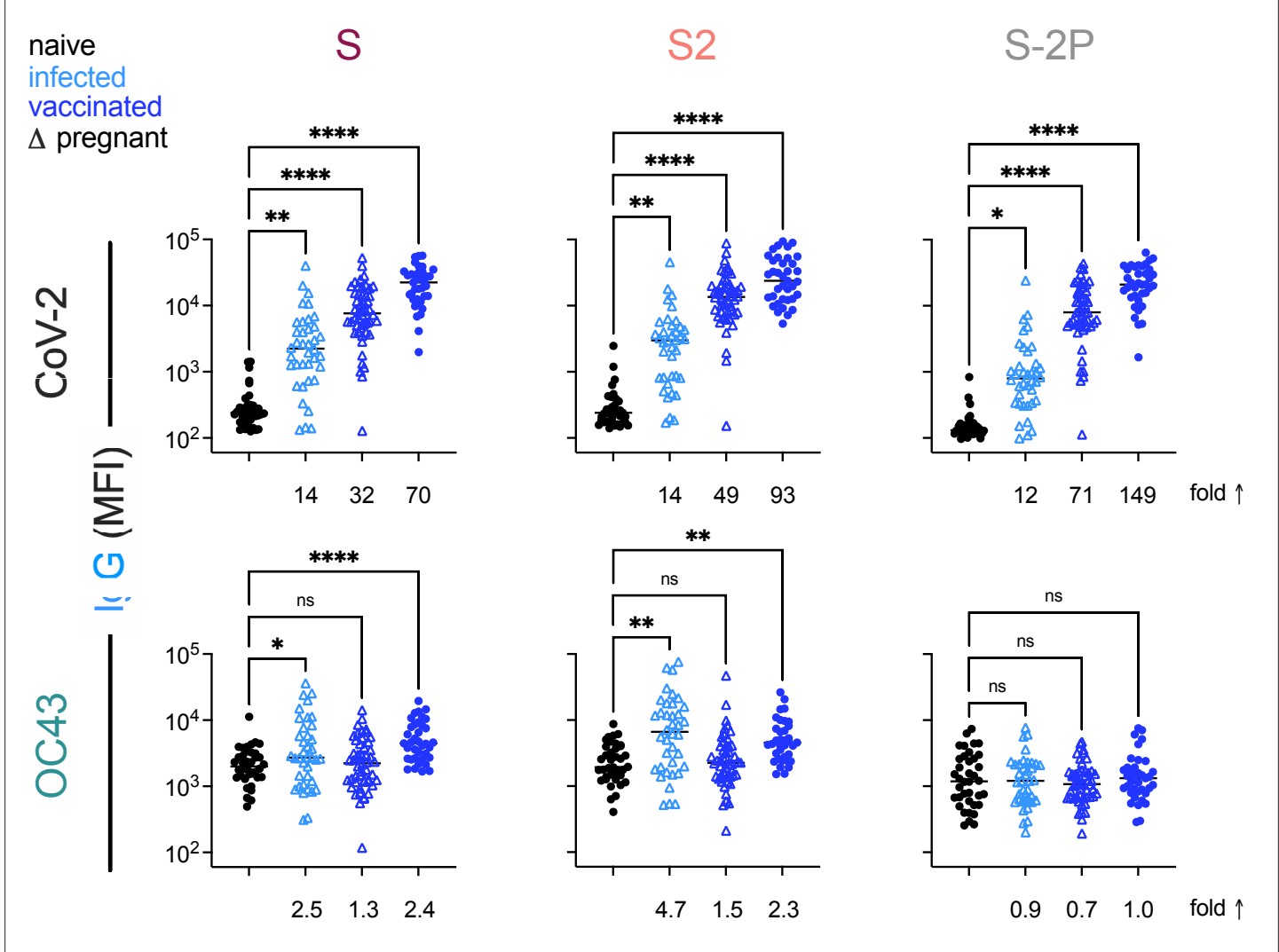

**Figure 7.** This Vaccination with stabilized spike does not result in robust boosting of endemic CoV responses. IgG responses to SARS-CoV-2 (top) and OC43 (bottom) spike proteins. For each CoV strain, responses to S (left), the S2 domain (center), and stabilized S (S-2P, right) are shown. Responses in SARS-CoV-2-naïve subjects are indicated in black, SARS-CoV-2-infected subjects in light blue, and SARS-CoV-2-vaccinated (mRNA) subjects in dark blue. Pregnant subjects are indicated with triangles. Statistical significance by ANOVA with Dunnett's correction (*p<0.05, **p<0.005, ****p<0.0001). Fold changes between mean response levels in seropositive and naïve cohorts are shown below each graph.

The online version of this article includes the following figure supplement(s) for figure 7:

**Figure supplement 1.** IgA and IgM responses among naïve subjects, infected pregnant women, vaccinated adults, and vaccinated pregnant women.

**Figure supplement 2.** Serology profile of contemporaneous commercial control subjects.

elevated levels of OC43-reactive IgA responses were not observed among vaccinated mothers, and neither infected nor vaccinated mothers showed elevated levels of OC43-reactive IgM (*Figure 7— figure supplement 1*). These differing immunogenicity profiles suggest that the presentation of native spike in the context of natural infection and stabilized spike in the context of mRNA vaccination are distinct. Lastly, to confirm this observation, responses, a validation cohort of mRNA-vaccinated healthy adults was evaluated (*Table 1*). In this cohort, while IgG responses to OC43 were statistically significantly elevated for S and S2 antigens, the effect was small (<2.5-fold) (*Figure 7*); elevated IgA or IgM responses were not observed (*Figure 7—figure supplement 1*).

In sum, elevated IgA and IgG but not IgM responses to the endemic CoV OC43 were observed in five distinct cohorts, including acutely infected, cross-sectional, and longitudinal convalescent cohorts. In contrast, two cohorts of subjects immunized with stabilized spike in the form of mRNA-based

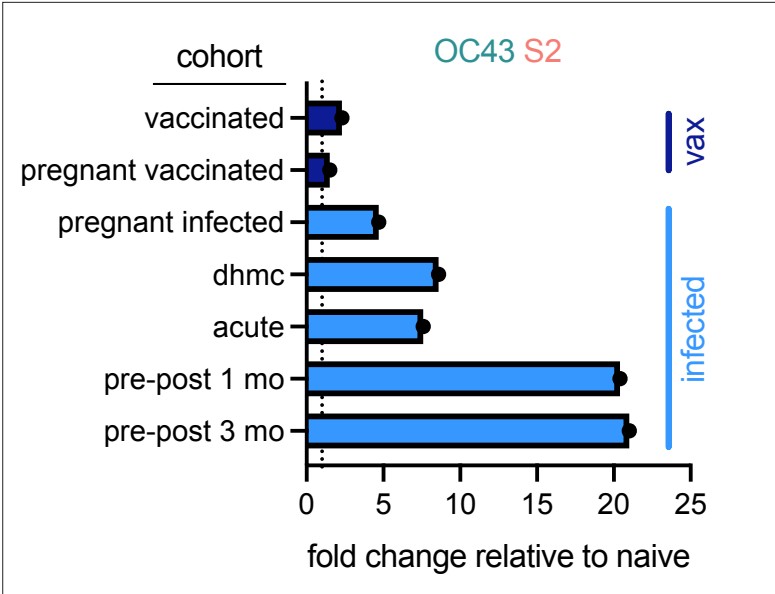

**Figure 8.** Differential boosting of responses to OC43 S2 by vaccination and natural infection. Mean fold changes in the magnitude of the IgG response to OC43 S2 between the indicated cohort and naïve control subjects, or median fold change between pre- (naïve) and post-infection timepoints (pre-post cohort at 1 and 3 months) for the longitudinal cohort. Dotted line indicates no change. Vaccinated (vax) and infected cohorts are indicated in dark and light blue, respectively.

vaccines showed no (IgA and IgM) or only a small (IgG only, and in only one cohort) elevation of binding to endemic CoV OC43 (*Figure 8*). In combination with the lack of boosting of responses that cross-react with endemic CoV, the high levels of neutralization activity observed to result from vaccination (*Jackson et al., 2020*) suggest favorable antigenicity of the prefusion conformation of S, and that, if detrimental, the costs of original antigenic sin might be avoided by immunogen design.

## Discussion

Cross-reactivity of endemic CoV antibodies against SARS-CoV-2 is of potential clinical relevance. The original antigenic sin hypothesis suggests that preexisting immunity results in the reactivation of a response to an earlier strain, as opposed to the formation of an unbiased response against the current strain. While this recall could offer temporal advantages, it may reduce the formation of neutralizing antibodies, thereby dampening effective clearance of the novel virus, as has recently been reported in mouse models (*Lapp et al., 2021*; *Lin et al., 2022*). Seminal studies of responses to influenza, based on epidemiology, modeling, and repertoire profiling, suggest that the antibodies generated from childhood exposure to influenza are 'imprinted' and exert a major influence on the nature of the antibody response elicited upon subsequent exposures in humans (*Francis et al., 1947*; *Fonville et al., 2014*; *Kucharski et al., 2015*; *Gostic et al., 2016*; *Lee et al., 2019*). It has been suggested that this phenomenon may exist in SARS-CoV-2 infection (*Aydillo et al., 2021*).

To better understand the role that prior exposure to CoV plays in antibody responses to SARS-CoV-2, we examined antigen-specific antibody responses against endemic and pandemic human CoV in subjects 1 month after SARS-CoV-2 infection. We measured heightened OC43-specific responses in SARS-CoV-2 convalescent subjects when compared to naïve controls and observed that the magnitude of these responses correlated with responses to SARS-CoV-2 spike protein and the S2 subdomain. Separately, we observed OC43-specific IgG, but not IgM, in acutely infected subjects 2 weeks post-positive COVID-19 diagnosis, indicating that these elevated responses were not likely to result from newly induced antibody lineages with cross-reactivity to OC43. Testing of serum samples pre- and post-SARS-CoV-2 infection reinforced the hypothesis that preexisting clonal families were boosted by the SARS-CoV-2 infection.

By isolating antigen-specific antibodies from convalescent serum, we determined that antibodies targeting SARS-CoV-2 S-2P and S2, but not the RBD, are cross-reactive with the spike of OC43. Similarly, antibodies isolated using OC43 were cross-reactive with SARS-CoV-2 S-2P and S2 but not the RBD. These results indicate that the cross-reactivity between these viral spike proteins is principally linked to the better-conserved S2 domain. Consistent with the observation that neutralizing antibodies tend to target the RBD of SARS-CoV-2 (*Ju et al., 2020*), we determined that both OC43 S-specific and SARS-CoV-2 S2-specific antibodies were non-neutralizing.

Over the course of this work, differing degrees of cross-reactivity were observed depending on whether prefusion-stabilized or unstabilized forms of S were used. In a comparison of naturally infected and vaccinated subjects representing exposure to unstabilized and proline-stabilized S antigens, respectively, the immunogens starkly contrasted. The immune response that resulted from vaccination with SARS-CoV-2 S protein stabilized in the prefusion conformation did not share the degree of cross-reactivity observed in those recovering from COVID-19. Reciprocally, elevated responses to stabilized OC43 were not observed in either infection or vaccination cohorts. Collectively, this data suggests both the critical importance of the conformational state of S to resulting humoral responses, but also the prevalence of epitopes that are not shared between pre- and postfusion or other conformations.

Whereas numerous studies have focused on the antibody responses elicited from vaccination toward the SARS-CoV-2 S protein and specifically the RBD, little is known about the effect previous exposure to endemic CoV may have on vaccination. However, our observations of limited boosting in the context of vaccination agree with other recent reports (*Skelly et al., 2021*; *Amanat et al., 2021*). Evaluation of a cohort vaccinated with a nonstabilized form of S will be required to begin to elucidate whether this is a feature of the stabilization itself, the mRNA-based vaccine modality, or if some other aspect of the vaccine formulation is responsible. Finally, this evidence of cross-reactivity has implications for the design of vaccines targeting SARS-CoV-2 variants. There is a risk that subsequent antibody responses might disregard the novel epitopes in favor of those already more familiar, although it should be emphasized that cross-reactivity was not a deterrent to the development of multiple highly efficacious SARS-CoV-2 vaccines.

Limitations of this study include the use of several small cohorts, some collected from distinct geographic locations, from subjects of varying age, who experienced differing disease severity, with collection at disparate time intervals from infection/vaccination, and who were not confirmed to lack a recent exposure to endemic CoV. While all natural infection cohorts showed evidence of boosting toward the endemic CoV OC43, it would be beneficial to survey boosting in cohorts that better span a range of disease severity and ages to examine whether those variables impact the observed boosting effect and outcomes of infection. To the extent that question has been investigated, it appears that boosting of responses to endemic CoV may be associated with poorer responses to SARS-CoV-2 (*Aydillo et al., 2021*).

Likewise, while both vaccination cohorts showed an absence of or reduction in boosting of cross-reactive responses, whether this absence is beneficial or detrimental cannot be determined from the data presented here. While some studies of natural infection have suggested that these boosted responses are not associated with protection, but instead with more severe disease (*Guo et al., 2021*; *Aydillo et al., 2021*; *Dugan et al., 2021*; *Lin et al., 2022*; *Anderson et al., 2021*; *Gombar et al., 2021*), others have supported the opposite conclusion (*Aran et al., 2020*; *Dugas et al., 2021a*; *Dugas et al., 2021b*). Collectively, these natural infection studies are challenged by the inability to distinguish between correlation and causation, and to separate contributions from humoral from cellular memory. While cross-reactive antibodies evaluated in this and other studies were non-neutralizing, neutralization is not the sole mechanism by which antibodies confer protection. Cross-reactive antibodies could interact with Fc receptors found on the surface of innate immune cells and promote protective effector function activities, including antibody-dependent cellular phagocytosis and antibody-dependent cellular cytotoxicity (*Fox et al., 2019*; *Earnest et al., 2019*; *Clark, 1997*). To this end, a subset of monoclonal antibodies isolated from SARS-CoV patients that were able to cross-react with, but not neutralize, SARS-CoV-2 were able to confer protection or reduce viral spread in mouse models (*Shiakolas et al., 2021*; *Beaudoin-Bussières et al., 2022*). Indeed, both passive transfer experiments of monoclonal antibodies (*Atyeo et al., 2021*; *Suryadevara et al., 2021*; *Tortorici et al., 2020*; *Winkler et al., 2021*; *Chan et al., 2021*; *Schäfer et al., 2021*) and evaluation of polyclonal antibodies raised in the context of vaccination or infection (*Alter et al., 2021*; *Francica*

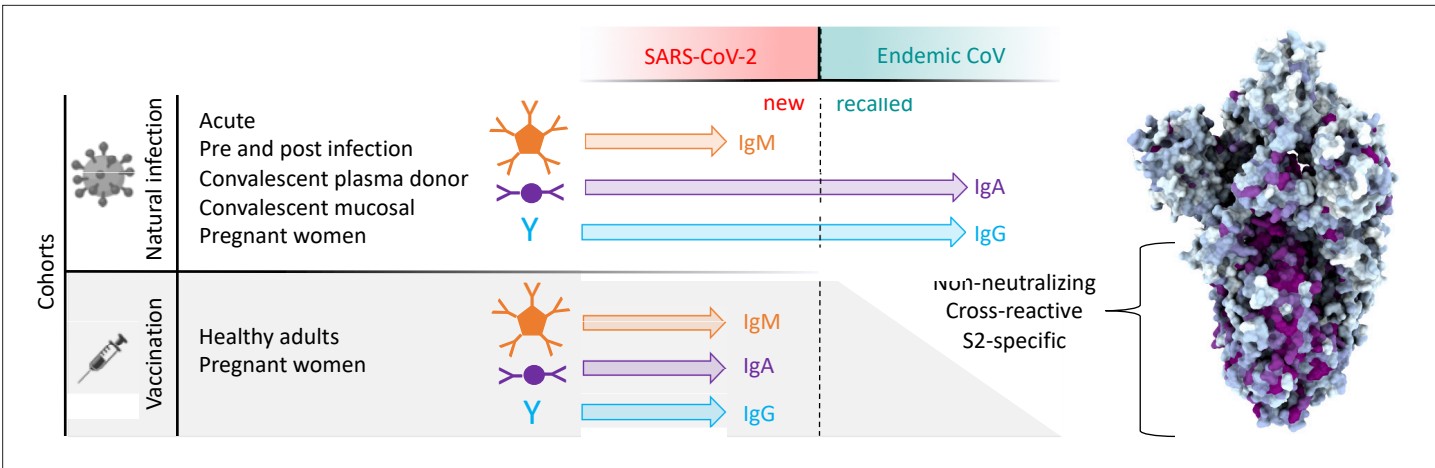

**Figure 9.** Graphical study summary. Antibody responses to SARS-CoV-2 and endemic CoV spike proteins were measured in diverse cohorts. While antibodies to SARS-CoV-2 were induced across all isotypes, only IgA and IgG responses to endemic CoV were robustly boosted, and only among naturally infected but not vaccinated individuals. These recalled, cross-reactive responses to endemic CoV primarily recognized the better-conserved S2 domain and were non-neutralizing. While other antiviral activities of broadly cross-reactive S2-specifc antibodies are not known, the differing antigenicity of natural infection and vaccination with stabilized prefusion spike has potential implications for the breadth and level of protection afforded by each.

*et al., 2021*; *Adeniji et al., 2021*) have shown that effector mechanisms contribute to antiviral activity in vivo. This evidence extends beyond correlative observations (*Tortorici et al., 2020*; *Alter et al., 2021*) to include mechanistic (*Peeri et al., 2020*) evidence of in vivo contributions via Fc sequence engineering to knock out or enhance these activities (*Suryadevara et al., 2021*; *Winkler et al., 2021*; *Yamin et al., 2021*), as well as on the basis of depletion of effector cells (*Winkler et al., 2021*). A full understanding of the contributions of effector activities of cross-reactive antibodies in the context of polyclonal responses in humans in general is also complicated by the likelihood that different S2 structural states may exist and different epitopes may be presented in the context of virions and infected cells, or among different tissue sites.

In sum, this study provides evidence that antibodies targeting OC43 are robustly boosted in response to SARS-CoV-2 infection but not vaccination with stabilized S, and that the S2 subdomain of the spike protein is likely responsible for triggering a recalled, IgG-dominated response (*Figure 9*). While non-neutralizing, the role of these cross-reactive antibodies in the context of infection is not yet known. Further work aimed at characterizing the effector function potential of these antibodies and understanding their role in vaccine-mediated protection could provide a more complete picture of the relative risks and benefits of this recall response relevant to vaccine design, particularly in the context of viral variants of concern.

## Materials and methods

**Key resources table**

| Reagent type (species) or resource | Designation | Source or reference | Identifiers | Additional information |
|---|---|---|---|---|
| Peptide, recombinant protein | H1N1 HA1 | Immune Technology | IT-003-00110p | |
| Peptide, recombinant protein | HSV gE | Immune Technology | IT-005-005p | |
| Peptide, recombinant protein | Tetanus toxoid | Sigma | 676570-37-9 | |
| Peptide, recombinant protein | SARS-CoV-2 N | Immune Technology | IT-002-033Ep | |
| Peptide, recombinant protein | SARS-CoV-2 fusion peptide | New England Peptide | | LCBiot-SKPSKRSFIEDLLFNK VTLADAGFIKQYGD |

*Continued on next page*

*Continued*

| Reagent type (species) or resource | Designation | Source or reference | Identifiers | Additional information |
|---|---|---|---|---|
| Peptide, recombinant protein | SARS-CoV-2 S1 | ACROBiosystems | S1N-C52H3-100ug | |
| Peptide, recombinant protein | SARS-CoV-2 RBD | BEI Resources | NR-52366 | |
| Peptide, recombinant protein | SARS-CoV-2 S2 | Immune Technology | IT-002-034p | |
| Peptide, recombinant protein | SARS-CoV-2 S-2P | *Wrapp et al., 2020* | | Produced in Expi 293 |
| Peptide, recombinant protein | SARS-CoV-2 S-6P | *Hsieh et al., 2020* | | Produced in Expi 293 |
| Peptide, recombinant protein | WIV1 S-2P | Plasmid provided by Jason McLellan | | Produced in Expi 293 |
| Peptide, recombinant protein | SARS-CoV-1 S | Sino Biological | 40634-V08B | |
| Peptide, recombinant protein | SARS-CoV-1 S1 | Sino Biological | 40150-V08B1 | |
| Peptide, recombinant protein | MERS S | Sino Biological | 40069-V08B | |
| Peptide, recombinant protein | MERS S1 | Sino Biological | 40069-V08B1 | |
| Peptide, recombinant protein | OC43 S | Sino Biological | 40607-V08B | |
| Peptide, recombinant protein | OC43 S-2P | Plasmid provided by Jason McLellan | | Produced in HEK 293F |
| Peptide, recombinant protein | OC43 S2 | Sino Biological | 40069-V08B | |
| Peptide, recombinant protein | 229E S | Sino Biological | 40601-V08H | |
| Peptide, recombinant protein | 229E S1 | Sino Biological | 40605-V08H | |
| Peptide, recombinant protein | HKU1 S | Sino Biological | 40606-V08H | |
| Peptide, recombinant protein | HKU1 S1 | Sino Biological | 40606-V08H | |
| Peptide, recombinant protein | NL63 S | Sino Biological | 40606-V08B | |
| Peptide, recombinant protein | NL63 S1 | Sino Biological | 40604-V08H | |
| Antibody | Anti-human IgG Fc-PE (goat polyclonal) | SouthernBiotech | 1030-09 | Used at 0.65 µg/mL. 40 µL used per well in 384-well plates |
| Antibody | Anti-human IgG1 Fc-PE (mouse monoclonal) | SouthernBiotech | 9054-09 | Used at 0.65 µg/mL. 40 µL used per well in 384-well plates |
| Antibody | Anti-human IgG2 Fc-PE (mouse monoclonal) | SouthernBiotech | 9070-09 | Used at 0.65 µg/mL. 40 µL used per well in 384-well plates |
| Antibody | Anti-human IgG3 Fc-PE (mouse monoclonal) | SouthernBiotech | 9210-09 | Used at 0.65 µg/mL. 40 µL used per well in 384-well plates |
| Antibody | Anti-human IgG4 Fc-PE (mouse monoclonal) | SouthernBiotech | 9200-09 | Used at 0.65 µg/mL. 40 µL used per well in 384-well plates |

*Continued on next page*

*Continued*

| Reagent type (species) or resource | Designation | Source or reference | Identifiers | Additional information |
|---|---|---|---|---|
| Antibody | Anti-human IgA Fc-PE (goat polyclonal) | SouthernBiotech | 2050-09 | Used at 0.65 µg/mL. 40 µL used per well in 384-well plates |
| Antibody | Anti-human IgA1 Fc-PE (mouse monoclonal) | SouthernBiotech | 9130-09 | Used at 0.65 µg/mL. 40 µL used per well in 384-well plates |
| Antibody | Anti-human IgA2 Fc-PE (mouse monoclonal) | SouthernBiotech | 9140-09 | Used at 0.65 µg/mL. 40 µL used per well in 384-well plates |
| Antibody | Anti-human IgM Fc-PE (mouse monoclonal) | SouthernBiotech | 9020-09 | Used at 0.65 µg/mL. 40 µL used per well in 384-well plates |
| Antibody | Anti-human IgM mouse/bovine/horse SP ads-UNLB (goat polyclonal) | SouthernBiotech | 2023-01 | |
| Antibody | Fab2 anti-human Fab2 (min × abs) (goat polyclonal) | Jackson Laboratories | 109-006-097 | |
| Antibody | Anti-human IgA (goat polyclonal) | SouthernBiotech | 2053-01 | |
| Cell line (*Homo sapiens*) | HEK Freestyle 293F | Thermo Fisher | R79007 | |
| Cell line (*H. sapiens*) | Expi293F | Thermo Fisher | A14527 | |
| Cell line (*H. sapiens*) | 293T-ACE2 | Takara | 631289 | |
| Recombinant DNA reagent | VSV-SARS-CoV-2 | *Letko et al., 2020* | | |
| Commercial assay or kit | Dynabeads | Thermo Fisher | 65011 | |
| Chemical compound, drug | EDC | Thermo Fisher | 22980 | |
| Chemical compound, drug | Sulfo-NHS | Thermo Fisher | 24510 | |
| Software, algorithm | GraphPad Prism version 9 | GraphPad Prism version 9 | | |
| Software, algorithm | ChimeraX | ChimeraX | | |
| Software, algorithm | R version 3.6.1 | R | | Packages: pracma, ggplot2, pheatmap |

## Structure visualization and manipulation

The sequence alignments were performed using Geneious 2021.1.1. Sequences and coordinates for CoV spike proteins were retrieved from the Protein Data Bank (PDB) entries SARS-CoV-2 (PDB 6XKL), 229E (PDB 6U7H), OC43 (PDB 6OHW), NL63 (PDB 5SZS), HKU1 (PDB 5I08), and SARS-CoV-2 Closed (PDB 6X6P). SARS-CoV-2 was structurally aligned to the other models by domain using the MatchMaker function with default parameters and visualized using Chimera version 1.15 (*Pettersen et al., 2004*). For structural characterization of conservation among the 60 complete genomes of the Coronaviridae suborder (https://www.ncbi.nlm.nih.gov/genomes/GenomesGroup.cgi?taxid=11118), Batch Entrez was used to find 585 associated proteins, which were then further downselected to spike proteins (N = 56) and aligned using Clustal Omega. This alignment was used to render by conservation and visualized using ChimeraX version 1.2 (*Goddard et al., 2018*).

## Human subjects

Initial study cohorts comprised 126 adult subjects interested in donating COVID-19 convalescent plasma. All were diagnosed with SARS-CoV-2 infection by PCR-based assays of nasopharyngeal swab, met the standard eligibility criteria for blood donation, and were collected in the Baltimore, MD, and Washington DC area (JHMI cohort), as previously described (*Klein et al., 2020*) and partially reported in *Natarajan et al., 2021*, and 26 SARS-CoV-2 convalescent individuals from the Lebanon, New Hampshire area (DHMC cohort) and partially reported previously in *Butler, 2020*. SARS-CoV-2

infection status was confirmed in all subjects by nasopharyngeal swab PCR. Plasma (JHMI) or serum (DHMC) was collected approximately 1 month after symptom onset or first positive PCR test.

The acute infection cohort comprised 10 subjects under the age of 30 with mostly asymptomatic infections diagnosed by positive results of PCR swabs in twice-weekly screening tests. Serum samples were collected 2 weeks following a positive test result. Three subsequently infected subjects over the age of 60 with mild symptoms for which pre-pandemic serum samples had been banked were evaluated pre-infection and at 2 and 10 weeks post-infection were also evaluated.

To support comparisons of natural infection and vaccination, two cohorts of pregnant women were evaluated. The first was a cohort of 38 subjects collected in Belgium who tested positive for SARS-CoV-2 in the third trimester of pregnancy. The second was a cohort of 50 subjects who were vaccinated with the Pfizer/BioNTech (BNT162) SARS-CoV-2 vaccine in the third trimester of pregnancy in Israel. Maternal serum samples were collected at the time of delivery.

A final cohort of 37 healthy adults vaccinated with Pfizer/BioNTech (BNT162) (n = 35) or Moderna (n = 2) vaccines whose serum was collected approximately 1 week after the second vaccine dose was evaluated to confirm observations of the differing immunogenicity observed among pregnant vaccine recipients.

Negative controls included samples from 15 naïve subjects collected from the Hanover, New Hampshire area, and from 38 subjects collected between April 2020 and January 2021 with negative results by serology (*Figure 7—figure supplement 2*) by a commercial vendor (BioIVT).

Human subject research was approved by the Johns Hopkins University School of Medicine's Institutional Review Board, the Dartmouth-Hitchcock Medical Center, CHU St. Pierre, Hadassah Medical Center, and BioIVT clinical site Committees for the Protection of Human Subjects as described in *Table 1*. Participants provided informed written consent. *Table 1* provides the basic clinical and demographic information for each cohort.

## Fc array assay

SARS-CoV-2 antigens, including spike protein in its trimeric and subdomain forms (i.e., S1, S2, RBD), endemic CoV, and the control antigens influenza HA and tetanus toxoid were covalently coupled to Luminex Magplex magnetic microspheres by two-step carbodiimide chemistry as previously described (*Brown et al., 2012*). Anti-isotype and subclass primary antibodies were used to quantify the total amount of each immunoglobulin isotype in a sample before (load) and after (eluate) antigen-specific antibody purification. The load was diluted in 1× PBS by 1:100 followed by seven fivefold serial dilutions. The eluate was diluted beginning from 1:10 with six fivefold serial dilutions. Antibody isotypes and subclasses were detected using R-phycoerythrin (PE)-conjugated secondary Abs as previously described (*Brown et al., 2017*). A FlexMap 3D array instrument was then used to measure the median fluorescence intensity (MFI) of the bead sets.

## Antigen-specific antibody purification

Human CoV antigens were covalently attached to magnetic Dynabeads (Thermo Fisher, 65011) using carbodiimide chemistry and as per the manufacturer's instructions. Briefly, 1 nmol of antigen was coupled to 300 µL of beads using a solution of 10 mg/mL EDC and 10 mg/mL sulfo-NHS. Bead activation and antigen coupling took place at room temperature with end-over-end mixing for 30 min and overnight, respectively. Following washes, the beads were reconstituted to 150 µL in PBS-TBN (Teknova, P0220).

The initial DHMC and JHMI cohorts were downselected to a total of 30 subjects, 15 from each cohort, representing a range of responses. Briefly, the self-reported case severities provided by the subjects of the DHMC cohort were used to select four mild, five moderate, and six severe subjects' samples. From the JHMI cohort, the subjects were ranked by their anti-spike titers and the five subjects closest to the 25th, 50th, and 75th percentiles were selected for purification.

In a nonbinding, clear-bottom 96-well plate, 5 µL of beads were diluted with 5 µL of 1× PBS before adding 50 µL of serum or plasma to the well. The plate was covered and shook at 800 rpm for 2.5 hr at room temperature. Using a magnetic base insert from a plate washer, the beads were pulled down from suspension for 1 min before decanting the waste. The beads were washed three times using 100 µL of PBS-TBN and 3 min of shaking for each wash. Following the third wash, the magnetic separation and decanting step was followed up by pipetting all residual buffer out of the wells while the

plate sat on the magnetic base. Antibodies specific to the antigen found on the beads were eluted using 20 µL of 1% formic acid (pH 2.9). After incubating with shaking for 10 min at room temperature and separating the beads, the eluate was pipetted from each well and transferred to a plate with wells containing 8 µL of 0.5 M sodium phosphate.

## Neutralization assay

Neutralization was performed using a VSV-SARS-CoV-2 pseudovirus assay as previously described (*Butler, 2020*; *Letko et al., 2020*). Briefly, serum or plasma samples were serially diluted twofold starting from a 1:25 dilution and incubated at 37°C for 1 hr with VSV-SARS-CoV-2 pseudovirus. Virus-serum/plasma mixtures were then added to pre-plated 293T-ACE2-expressing target cells in white 96-well plates at a final volume of 100 µL per well and incubated at 37°C for 24 hr. To test for neutralization by antibody eluates, samples were first concentrated 10-fold using Amicon Ultra 0.5 mL centrifugal filter devices (molecular weight cutoff of 100K Da) followed by twofold serial dilution from 1:30, incubation with VSV-SARS-CoV-2 S pseudovirus, and addition of the mixtures to 293T-ACE2 target cells as described. Luciferase activity was measured using the Bright-Glo system and percent neutralization determined relative to control wells consisting of 293T-ACE2 cells infected with the pseudovirus alone.

## Data analysis

Basic statistical data analysis and visualization or raw Fc Array data were performed using GraphPad Prism, with statistical tests described in each figure legend. Heatmaps were visualized using the 'pheatmap' (*pheatmap, 2019*) package with hierarchical cluster analysis (*Bridges, 2016*) defined using Manhattan distance (*Minkowski, 1910*) in R version 3.6.1. Fc Array features were log transformed, then scaled and centered by their standard deviation from the mean (z-score).

In order to quantify the enrichment of antigen-specific antibodies, individual features (antigen-detection pair) were plotted relative to the total immunoglobulin isotype in serum across a titration range for each sample. A generalized additive model (GAM) (*Hastie and Tibshirani, 2017*) with a cubic spline basis was used to fit a smoothed curve to load and eluate sample data using the 'ggplot2' package in R (*Wickham, 2016*). To quantify the difference between the load and eluate curves, we fit a separate GAM curve to the differences of the predicted values from the load and eluate GAM curves over total antibody as previously suggested (*Rose et al., 2012*). We approximated the area under the difference curve (AUC) using the trapezoidal rule (*Whittaker, 1967*; *Poisson, 1827*) in the 'pracma' (*Borchers, 2021*) package in R.

## Data availability

Data that support the findings of this study are available at https://github.com/AckermanLab/Crowley_et_al_SARS-CoV-2_Boosting, (copy archived at swh:1:rev:25dbaa42711c91ca06661794e8e11a-b72eacdbae; *AckermanLab, 2021*).

# Acknowledgements

We acknowledge the Immune Monitoring and Flow Cytometry Resource (IMFCSR) at the Norris Cotton Cancer Center at Dartmouth supported by NCI Cancer Center Support Grant 5P30 CA023108-41. We thank all the participants who enrolled and laboratory staff who helped collect and process the samples. VSV pseudovirus expression plasmids were provided by Michael Letko (Rocky Mountain Laboratories), CoV S-2P, S-6P, and RBD-Fc expression constructs were provided by Jason McLellan (University of Texas at Austin), and fusion peptide was provided by Laura Walker and Mrunal Sakharkar (Adimab). The following reagent was produced under HHSN272201400008C and obtained through BEI Resources, NIAID, NIH: Spike Glycoprotein Receptor-binding domain (RBD) from SARS-Related Coronavirus 2, Wuhan-Hu-1 with C-Terminal Histidine Tag, Recombinant from Baculovirus, NR-52307. The following reagent was deposited by the Centers for Disease Control and Prevention and obtained through BEI Resources, NIAID, NIH: SARS Related Coronavirus 2, Isolate USA-WA1/2020, NR-52281. This work was supported in part by the Division of Intramural Research, National Institute of Allergy and Infectious Diseases, as well as extramural support from the National Institute of Allergy and Infectious Diseases (U19AI145825 to

MEA, R01AI120938, R01AI120938S1, and R01AI128779 to AART), National Heart Lung and Blood Institute (K23HL151826 to EMB), National Institute of General Medical Sciences (P20-GM113132 BioMT Molecular Tools Core).

## Additional information

### Funding

| Funder | Grant reference number | Author |
|---|---|---|
| National Institutes of Health | U19AI145825 | Margaret E Ackerman |
| National Institutes of Health | R01AI120938 | Aaron AR Tobian |
| National Institutes of Health | R01AI120938S1 | Aaron AR Tobian |
| National Institutes of Health | R01AI128779 | Aaron AR Tobian |
| National Institutes of Health | intramural | Andrew D Redd |
| National Institutes of Health | K23HL151826 | Evan M Bloch |
| National Institutes of Health | P20-GM113132 | Margaret E Ackerman |

The funders had no role in study design, data collection and interpretation, or the decision to submit the work for publication.

### Author contributions

Andrew R Crowley, Harini Natarajan, Formal analysis, Investigation, Methodology, Visualization, Writing - original draft, Writing – review and editing; Andrew P Hederman, Investigation, Visualization, Writing - original draft, Writing – review and editing; Carly A Bobak, Formal analysis, Software, Writing – review and editing; Joshua A Weiner, Data curation, Supervision, Visualization, Writing – review and editing; Wendy Wieland-Alter, Investigation, Resources, Writing – review and editing; Jiwon Lee, Evan M Bloch, Aaron AR Tobian, Andrew D Redd, Joel N Blankson, Dana Wolf, Tessa Goetghebuer, Arnaud Marchant, Resources, Writing – review and editing; Ruth I Connor, Investigation, Methodology, Writing – review and editing; Peter F Wright, Resources, Supervision, Writing – review and editing; Margaret E Ackerman, Conceptualization, Supervision, Writing - original draft, Writing – review and editing

### Author ORCIDs

Margaret E Ackerman http://orcid.org/0000-0002-4253-3476

### Ethics

Human subject research was approved by the Johns Hopkins University School of Medicine's Institutional Review Board, the Dartmouth-Hitchcock Medical Center, CHU St. Pierre, Hadassah Medical Center, and BioIVT clinical site Committees for the Protection of Human Subjects as described in Table 1. Participants provided informed written consent.

### Decision letter and Author response

Decision letter https://doi.org/10.7554/eLife.75228.sa1
Author response https://doi.org/10.7554/eLife.75228.sa2

## Additional files

### Supplementary files

• Transparent reporting form

## Data availability

Source data and code for a subset of cohorts and study data can be found at: https://github.com/AckermanLab/Butler_et_al_COVID_2020, (copy archived at swh:1:rev:2765721adeeef5ca2db5e9b9d-e70d1bd5e6c50a9) https://github.com/AckermanLab/Natarajan_et_al_COVID_2021, (copy archived at swh:1:rev:cc8bdc2cc771587aad33a17b0fd96dd0c63f238a).

The following previously published datasets were used:

| Author(s) | Year | Dataset title | Dataset URL | Database and Identifier |
|---|---|---|---|---|
| Wrapp D, Hsieh C-L, Goldsmith JA, McLellan JS | 2020 | SARS-CoV-2 HexaPro S One RBD up | https://www.rcsb.org/structure/6XKL | RCSB Protein Data Bank, 6XKL |
| Li Z, Benlekbir S, Rubinstein JL, Rini JM | 2019 | Cryo-EM structure of the HCoV-229E spike glycoprotein | https://www.rcsb.org/structure/6U7H | RCSB Protein Data Bank, 6U7H |
| Tortorici MA, Walls AC, Lang Y, Wang C, Li Z, Koerhuis D, Boons GJ, Bosch BJ, Rey FA, de Groot R, Veesler D | 2019 | Structural basis for human coronavirus attachment to sialic acid receptors. Apo-HCoV-OC43 S | https://www.rcsb.org/structure/6OHW | RCSB Protein Data Bank, 6OHW |
| Walls AC, Tortorici MA, Frenz B, Snijder J, Li W, Rey FA, DiMaio F, Bosch BJ, Veesler D | 2016 | Glycan shield and epitope masking of a coronavirus spike protein observed by cryo-electron microscopy | https://www.rcsb.org/structure/5SZS | RCSB Protein Data Bank, 5SZS |
| Kirchdoerfer RN, Cottrell CA, Wang N, Pallesen J, Yassine HM, Turner HL, Corbett KS, Graham BS, McLellan JS, Ward AB | 2016 | Prefusion structure of a human coronavirus spike protein | https://www.rcsb.org/structure/5I08 | RCSB Protein Data Bank, 5I08 |
| Herrera NG, Morano NC, Celikgil A, Georgiev GI, Malonis R, Lee JH, Tong K, Vergnolle O, Massimi A, Yen LY, Noble AJ, Kopylov M, Bonanno JB, Garrett-Thompson SC, Hayes DB, Brenowitz M, Garforth SJ, Eng ET, Lai JR, Almo SC | 2020 | Characterization of the SARS-CoV-2 S Protein: Biophysical, Biochemical, Structural, and Antigenic Analysis | https://www.rcsb.org/structure/6X6P | RCSB Protein Data Bank, 6X6P |

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
