## [Editor Report]

This study is aimed to determine whether infection or vaccination affects activation of preexisting memory B cells. The data are clear and have implications for further development of a new generation of vaccines.

---

## [Decision Letter]

**Decision letter after peer review:**

Thank you for submitting your article "Boosting of cross-reactive antibodies to endemic coronaviruses by SARS-CoV-2 infection but not vaccination with stabilized spike" for consideration by *eLife*. Your article has been reviewed by 3 peer reviewers, one of whom is a member of our Board of Reviewing Editors, and the evaluation has been overseen Tadatsugu Taniguchi as the Senior Editor. The reviewers have opted to remain anonymous.

Essential revisions:

1) It was confused by the data in Figure 7 where the pregnant vaccinated cohort showed undetectable level of cross-reactivity, which is understandable based on data in Figure 1-6, but the vaccinated cohort showed detectable levels of cross-reactivity against OC43. From the information provided in the method section, I think the pregnant cohort had a longer collection time after 2nd vaccination compared to the non-pregnant cohort which is collected 1 week after their 2nd shot. Based on this, I presume that vaccination with stabilized-spike induces acute, short-lived cross-reactivity against endemic CoV. Or is this due to CoV-2 infection history in the non-pregnant cohort?

2) The number of naïve samples is different in the main text (page 7 line 94) and the methods (page 13 line 377).

3) Please provide the time window of historical negative control sample collection.

4) For readers, authors had better to discuss the Fc-mediated vaccine effect of non-neutralizing antibodies especially on the point of how non-neutralizing antibodies, such as ones described in this manuscript targeting the S2 domain (mainly exposed in the denatured state), recognize the denatured form of spike proteins during infection and how the immune complex of those denatured spike proteins and antibodies can turn into in vivo effect of protection as described in the Discussion section by authors.

As seen in the below reviewers' comments, all reviewers are essentially positive for this manuscript. But, they have several concerns.

*Reviewer #1 (Recommendations for the authors):*

This is a nice series of serological careful analysis, although isolating single cell Igs, and transfer of cross-reactive Abs into SARS-CoV2-infected mice are desirable. Nevertheless, this study have added the proof-of-concept that novel pandemic SARS-CoV-2 viruses, like influenza viruses, still can activate the pre-existing cross-reactive memory B cells.

*Reviewer #2 (Recommendations for the authors):*

1. I was confused by the data in Figure 7 where the pregnant vaccinated cohort showed undetectable level of cross-reactivity, which is understandable based on data in Figure 1-6, but the vaccinated cohort showed detectable levels of cross-reactivity against OC43. From the information provided in the method section, I think the pregnant cohort had a longer collection time after 2nd vaccination compared to the non-pregnant cohort which is collected 1 week after their 2nd shot. Based on this, I presume that vaccination with stabilized-spike induces acute, short-lived cross-reactivity against endemic CoV. Or is this due to CoV-2 infection history in the non-pregnant cohort?

2. The number of naïve samples is different in the main text (page 7 line 94) and the methods (page 13 line 377).

3. Please provide the time window of historical negative control sample collection.

*Reviewer #3 (Recommendations for the authors):*

Data presented in this manuscript using human samples are highly helpful for future vaccine studies. The authors carefully mention that the cross-reactive epitope which they found is not proved to be directly linked to the biological significance of the vaccine outcome. I have no major concern with both the data presentation and the manuscript. For readers, authors might want to discuss the Fc-mediated vaccine effect of non-neutralizing antibodies especially on the point of how non-neutralizing antibodies, such as ones described in this manuscript targeting the S2 domain (mainly exposed in the denatured state), recognize the denatured form of spike proteins during infection and how the immune complex of those denatured spike proteins and antibodies can turn into in vivo effect of protection as described in the Discussion section by authors.

---

## [Author Response]

Essential revisions:1) It was confused by the data in Figure 7 where the pregnant vaccinated cohort showed undetectable level of cross-reactivity, which is understandable based on data in Figure 1-6, but the vaccinated cohort showed detectable levels of cross-reactivity against OC43.

We agree that the vaccinated non-pregnant cohort exhibit statistically significantly elevated levels of IgG antibodies to OC43 S and S2. We note that this has also been observed in other studies (doi:10.1038/s41467-021-25167-5, doi:10.1016/j.cell.2021.06.005). We think it is important to focus on the magnitude of the observed boosting effect. To better emphasize these differences in magnitude, the revised manuscript now includes a new figure (revised manuscript Figure 8).

From the information provided in the method section, I think the pregnant cohort had a longer collection time after 2nd vaccination compared to the non-pregnant cohort which is collected 1 week after their 2nd shot. Based on this, I presume that vaccination with stabilized-spike induces acute, short-lived cross-reactivity against endemic CoV. Or is this due to CoV-2 infection history in the non-pregnant cohort?

The two vaccinated cohorts do have different sampling times post 2^nd^ immunization. The reviewer’s hypothesis that boosted responses toward endemic CoV may decay rapidly is consistent with the lower magnitude of boosting observed in this cohort at ~1 week than in the vaccinated pregnant cohort at ~3 weeks.

However, it is also possible that differences in the magnitude of boosting between vaccinated pregnant and non-pregnant cohorts relate to other differences (ie: demographics/geography/pregnancy status). It is also possible that this difference is not statistically significant. To address this latter possibility, IgG response magnitudes between pregnant and non-pregnant vaccine recipients are compared in Author response image 1. Based on this analysis, the non-pregnant vaccine recipients do exhibit significantly elevated responses to both CoV-2 and OC43 S and S2, as compared to pregnant vaccine recipients.

**Author response image 1. sa2fig1:** Comparison of IgG responses to CoV-2 and OC43 spike in vaccinated subjected. IgG responses to SARS CoV-2 (top) and OC43 (bottom) S (left), S2 (center), and S-2P (right), among pregnant (P) and non-pregnant (NP) vaccinated subjects (Vax). Statistical significance defined by two-sided Mann Whitney test with unadjusted p values reported in inset.

As to infection history, we are unable to formally exclude the possibility that prior CoV-2 infection influences the vaccinated non-pregnant cohort. Looking into the details of this cohort, we found that 4 of the 37 non-pregnant vaccine recipients had a prior history (timing unknown) of CoV-2 infection. These subjects do not appear to be distinct from the rest of the non-pregnant vaccine recipients in terms of their degree of boosting toward OC43 (Author response image 2) . They do appear to generally be among the higher responders in terms of CoV-2-specific responses.

**Author response image 2. sa2fig2:** CoV-2 and OC43-specific IgG responses among vaccinated subjects with and without prior infection. IgG responses to OC43 S (left), S2 (center), and S-2P (right), among naïve subjects and non-pregnant vaccinated subjects (Vax). Vaccinated subjects are colored by vaccine (green – Moderna; blue – Pfizer) or history of prior infection (red, N = 4, Pfizer).

It is also possible that other subjects among this group also had prior infection that was undiagnosed/unreported. In sum, it is difficult for us to support hypotheses about the influence of kinetics, prior infection history, or other factors. We have checked the manuscript for clarity about these possible influences, and have added differences in timing of sample collection as a specific limitation (line 310).

2) The number of naïve samples is different in the main text (page 7 line 94) and the methods (page 13 line 377).

This error in the methods section has been corrected (line 392).

3) Please provide the time window of historical negative control sample collection.

The methods section now provides the period (Apr 2020 – Jan 2021) during which controls were collected (lines 392-395). In checking on this set of samples, we realized that the vendor had not provided us with draw date information. We sourced this information, and found that the vendor did not provided us with “historical” controls. As such, we have amended labeling of this cohort. Because these samples were collected during the pandemic, we have evaluated seropositivity by comparison to vaccinated pregnant subjects in Figure 7—figure supplement 2. Hierarchical clustering of these samples clearly distinguishes them from the majority of the vaccinated subjects. We may have sufficient sample remaining to evaluate seronegative status using a clinical grade test if felt necessary by reviewers, but we propose that given the available serology data, more formal testing may present a significant expense with relatively little scientific value.

4) For readers, authors had better to discuss the Fc-mediated vaccine effect of non-neutralizing antibodies especially on the point of how non-neutralizing antibodies, such as ones described in this manuscript targeting the S2 domain (mainly exposed in the denatured state), recognize the denatured form of spike proteins during infection and how the immune complex of those denatured spike proteins and antibodies can turn into in vivo effect of protection as described in the Discussion section by authors.

The revised manuscript now more thoroughly discusses possible mechanisms whereby S2-specific antibodies may make positive contributions in vivo (lines 335-339).

Reviewer #2 (Recommendations for the authors):1. I was confused by the data in Figure 7 where the pregnant vaccinated cohort showed undetectable level of cross-reactivity, which is understandable based on data in Figure 1-6, but the vaccinated cohort showed detectable levels of cross-reactivity against OC43. From the information provided in the method section, I think the pregnant cohort had a longer collection time after 2nd vaccination compared to the non-pregnant cohort which is collected 1 week after their 2nd shot. Based on this, I presume that vaccination with stabilized-spike induces acute, short-lived cross-reactivity against endemic CoV. Or is this due to CoV-2 infection history in the non-pregnant cohort?

Addressed above.

2. The number of naïve samples is different in the main text (page 7 line 94) and the methods (page 13 line 377).

Corrected.

3. Please provide the time window of historical negative control sample collection.

Provided.

Reviewer #3 (Recommendations for the authors):Data presented in this manuscript using human samples are highly helpful for future vaccine studies. The authors carefully mention that the cross-reactive epitope which they found is not proved to be directly linked to the biological significance of the vaccine outcome. I have no major concern with both the data presentation and the manuscript. For readers, authors might want to discuss the Fc-mediated vaccine effect of non-neutralizing antibodies especially on the point of how non-neutralizing antibodies, such as ones described in this manuscript targeting the S2 domain (mainly exposed in the denatured state), recognize the denatured form of spike proteins during infection and how the immune complex of those denatured spike proteins and antibodies can turn into in vivo effect of protection as described in the Discussion section by authors.

Additional comments regarding mechanism have been added as described above.